# The Activity of Human NK Cells Towards 3D Heterotypic Cellular Tumor Model of Breast Cancer

**DOI:** 10.3390/cells14141039

**Published:** 2025-07-08

**Authors:** Anastasia Leonteva, Maria Abdurakhmanova, Maria Bogachek, Tatyana Belovezhets, Anna Yurina, Olga Troitskaya, Sergey Kulemzin, Vladimir Richter, Elena Kuligina, Anna Nushtaeva

**Affiliations:** 1Scientific Center of Genetics and Life Sciences, Sirius University of Science and Technology, 1 Olimpiysky Avenue, Krasnodar Region, Sirius 354340, Russia; anastleont@mail.ru (A.L.); maryambogachek@gmail.com (M.B.); 2Institute of Chemical Biology and Fundamental Medicine, Siberian Branch of the Russian Academy of Sciences, Akad. Lavrentiev Ave. 8, Novosibirsk 630090, Russia; m.abdurakhmanova98@gmail.com (M.A.); troitskaya_olga@bk.ru (O.T.); richter@niboch.nsc.ru (V.R.); kuligina@niboch.nsc.ru (E.K.); 3Institute of Molecular and Cellular Biology, Siberian Branch of the Russian Academy of Sciences, Akad. Lavrentiev Ave. 8/2, Novosibirsk 630090, Russia; ochotanya@gmail.com (T.B.); a.yurina@g.nsu.ru (A.Y.); skulemzin@mcb.nsc.ru (S.K.); 4Department of Natural Sciences, Novosibirsk State University, Pirogova Str. 2, Novosibirsk 630090, Russia

**Keywords:** 3D cancer cell models, spheroids, co-culture, patient-derived cell culture, tumor microenvironment, stromal cell, cancer-associated fibroblast, natural killer cell, immunotherapy, breast cancer

## Abstract

Due to the complexity of modeling tumor-host interactions within the tumor microenvironment in vitro, we developed a 3D heterotypic cellular breast cancer (BC) model. We generated spheroid models using MCF7, MDA-MB-231, and SK-BR-3 cell lines alongside cancer-associated (BrC4f) and normal (BN120f) fibroblasts in ultra-low attachment plates. Stromal spheroids (3Df) were formed using a liquid overlay technique (graphical abstract). The YT cell line and peripheral blood NK (PB-NK) cells were used as immune components in our 3D model. In this study, we showed that stromal cells promoted tumor cell aggregation into spheroids, regardless of the initial proliferation rates, with NK cells accumulating in fibroblast-rich regions. The presence of CAFs within the model induced alterations in the expression levels of MICA/B and PD-L1 by tumor cells within the 3D-2 model. The feasibility of utilizing a 3D cell BC model in combination with cytokines and PB-NKs was evaluated. We observed that IL-15 and IL-2 enhanced NK cell activity within spheroids, whereas TGFβ had varying effects on proliferation depending on the cell type. Stimulation with IL-2 and IL-15 or TGFβ1 altered PB-NK markers and stimulated their differentiation into ILC1-like cells in 3D models. These findings underscore the regulatory function of CAFs in shaping the response of the tumor microenvironment to immunotherapeutic interventions.

## 1. Introduction

In terms of global prevalence, breast cancer (BC) is the most common cancer diagnosed in women. Approximately 20% of patients with BC develop metastases in various organs, which are difficult to treat [1]. BC is a solid tumor with different biological and morphological properties that affect clinical outcome prediction and treatment [2]. A wide range of therapeutic modalities, including chemotherapy, immunotherapy, radiotherapy, and targeted therapies, are currently used to treat patients with BC. The choice of systemic therapy depends on the hormone status of breast cancer (ER and PR expression) and the expression of HER2, determined at diagnosis of the patient [3]. Simultaneously, interactions with immune and stromal cells in the tumor microenvironment (TME) may enable breast cancer (BC) cells to escape immune surveillance, ultimately leading to therapeutic resistance, recurrence, and metastasis [3,4]. Cancer-associated fibroblasts (CAF) are one of the most abundant components of the breast tumor microenvironment and are major contributors to immune modulation. CAFs regulate the activity of many immune cells, including T cells, macrophages, and dendritic cells; however, little is known about their interaction with NK cells, which constitute an important arm of antitumor immunity [5]. Furthermore, solid tumors create a specific microenvironment characterized by hypoxia, nutrient deprivation, waste accumulation, and pH gradients [6]. These factors may impair NK cell infiltration and cytotoxic function, ultimately altering the tumor susceptibility to NK cell-mediated killing [7,8]. It has been shown that the expression of NKp30, NKG2D, and CD16 receptors on the surface of NK cells is usually decreased at the late stages of the disease, resulting in lower activity of NK cells in the tumor. In contrast, the expression of inhibitory receptors, such as NKG2A, increases during cancer progression and is associated with the diminished cytotoxic activity of natural killer (NK) cells [9]. Factors secreted by stromal cells in the tumor microenvironment (e.g., TGFβ), as well as by immune and tumor cells (prostaglandin E2, adenosine), can impact NK cell functionality [9,10].

Type I cytokines, such as IL-2 and IL-15, are frequently used in cancer cell therapy to enhance NK cell function. IL-2 has been shown to be a potent driver of NK cell proliferation and has been observed to stimulate the production of lytic molecules [11]. IL-15 is a crucial cytokine that is indispensable for the development, survival, and proliferation of NK cells [12]. A low dose of IL-15 is sufficient to maintain survival signaling in NK cells, whereas a higher dose of IL-15 promotes NK cell proliferation and enhances the expression of effector molecules, such as perforin and granzymes [13]. However, the formation of an immunosuppressive microenvironment as a consequence of tumor immunity evasion has been shown to result in a decrease in the efficiency of NK cells in combating tumors through the secretion of suppressive cytokines [14]. For instance, the cytokine transforming growth factor beta (TGFβ) plays a pivotal role in suppressing NK cell migration, cytotoxic activity, and the production of cytokines, including interferon gamma (IFN-γ) and tumor necrosis factor (TNF) [14,15]. Chung et al., 2024 demonstrated that TGF-β1 and IL-15 induce NK cells to adopt immunosuppressive properties [16].

Robust and reproducible experimental models are vital for elucidating cellular interaction mechanisms and promoting the development of effective cancer therapies. Immortalized cell lines are routinely used in most cancer-related studies investigating the mechanisms of BC tumorigenesis and drug development. These in vitro 2D models provided important insights into potential anticancer agents at the early stages of therapeutic agent discovery; however, they do not represent the TME complexity and the gradients of nutrients, oxygen, and drug penetration occurring in vivo [17,18]. To address the shortcomings of traditional cancer 2D-models, over the last two decades there has been significant progress in the development of 3D cellular models over the last two decades [19,20,21].

3D cellular models can mimic the physiological properties of cells in terms of topographical and mechanical forces, rendering cellular responses to different stimuli [22,23,24]. In order to establish 3D cellular model, researchers use low-adhesion plates, the hanging drop method, cell-seeded matrices and scaffolds, micropatterning, and agitation-based methods [25]. Heterotypic cellular models have proven their efficacy in elucidating novel aspects of cancer biology [22,26,27]. A heterotypic spheroid can be defined as a 3D cell culture model composed of multiple cell types (two types, 3D-2, or three types, 3D-3), in contrast to homotypic spheroids (3D), which consist of a single cell type. Notably, heterotypic cellular 3D tumor models could be useful for studying the infiltration and activation of immune cells, making them a versatile in vivo-like model [22,28]. Studies employing in vitro heterotypic models from different tumor pathologies have also integrated stromal elements like cancer-associated fibroblasts and immune subtypes, such as macrophages and NK cells, to better recapitulate the tumor microenvironment [19,28]. However, many of these models include tumor and stromal cells originating from distinct tissue sites, such as colon adenocarcinoma cells and normal fetal lung fibroblasts [28], breast cancer cells, and human dermal fibroblasts [29], which cannot accurately reproduce the specific cell-to-cell interactions of this tissue in the studied organ.

The aim of the present study was to create a 3D heterotypic cellular BC model. The selection of tumor cellular models for simulating the three molecular types of breast cancer was based on the hormonal status of the cell lines. To assess the influence of hormone signaling, we selected three breast cancer cell lines representing different hormonal profiles: hormone-positive MCF-7 and HER2-positive SK-BR-3 cell lines, and the triple-negative MDA-MB-231 cell line. Two different types of patient-derived cultured fibroblasts originating from normal or tumor breast tissue of the patient were used to create spheroid models. The cell culture of BrC4f was categorized as CAFs, while the BN120f cell culture represented normal fibroblasts. PB-NK and YT cells are both immune cells, but have different origins and characteristics. PB-NK cells are a natural component of the innate immune system found in the blood, while YT is a cell line derived from a human T-cell leukemia tumor [30]. Both cell types exhibit cytotoxic activity; however, the primary function of YT cells is as a research tool, whereas PB-NK cells have been the subject of more extensive studies regarding therapeutic applications. Depending on whether normal fibroblasts or CAFs were used in a given 3D model, the following points were evaluated in the present study: (i) a comparison of NK cell killer activity on two-dimensional (2D) and three-dimensional (3D) cultures of tumor or stromal components of BC, (ii) the penetration of NK cells into the spheroid model, and (iii) the modulation of the immune response by stimulation with a combination of IL-2, IL-15, or TGF-β1.

## 2. Materials and Methods

### 2.1. Cell Lines

MCF-7 (#ACC 115, DSMZ, Braunschweig, Germany), MDA-MB-231 (#ACC 65, DSMZ, Braunschweig, Germany), SK-BR-3 (ATCC, #HTB-30, Manassas, VA, USA), and NK cell line YT (#ACC 434, DSMZ, Braunschweig, Germany) were purchased from the American Type Culture Collection (ATCC, Manassas, VA, USA), and BrC4f and BN120f were obtained from the Laboratory of Biotechnology at the Institute of Chemical Biology and Fundamental Medicine SB RAS (ICBFM, Novosibirsk, Russia) [31]. The BrC4f culture exhibited the characteristics of cancer-associated fibroblasts (CAFs) [30], while BN120f exhibited the characteristics of normal fibroblasts [26]. All cell lines were free of mycoplasma contamination, as determined by RT-PCR. Genetic identification of the cell lines was performed using the GOrDIS Plus kit (GORDIZ, Moscow, Russia). The STR profiles correspond to those published in the ATCC, DSMZ, and Cellosaurus international databases.

MCF-7, BrC4f, BrC120f, BN120f, and NK cell line YT cells were cultured in Iscove’s Modified Dulbecco’s Medium (IMDM) (#I7633-10X1L, Sigma-Aldrich, St. Louis, MO, USA), MDA-MB-231 cells were cultured in Dulbecco’s Modified Eagle’s Medium (DMEM) (#D6046-1L, Sigma-Aldrich, St. Louis, MO, USA), and SK-BR-3 cells were cultured in Dulbecco’s Modified Eagle Medium/Nutrient Mixture F-12 (DMEM/F12) (#42400028, Gibco™, New York, NY, USA). All culture media were supplemented with 10% fetal bovine serum (FBS) (#A316040, Thermo Fisher, Waltham, MA, USA), GlutaMAX™ solution (#35050061, Gibco™, New York, NY, USA), 250 mg/mL amphotericin B, and 100 U/mL penicillin/streptomycin (#15140122, Gibco™, Waltham, MA, USA). The culture media for SK-BR-3 cells was supplemented with MEM Non-Essential Amino Acids solution (#11140050, Gibco™, Waltham, MA, USA) and Sodium Pyruvate (#11360070, Gibco™, Waltham, MA, USA). Cells were cultured at 37 °C with 5% CO_2_ unless otherwise mentioned.

During passivation of adherent cultures, the cells were washed with 2 mL of 1× PBS and incubated at 37 °C for 3–5 min with 400 μL of TrypLE™ (#12604013, GIBCO, Invitrogen, Waltham, MA, USA) solution. After detachment, the cells were washed with 1 mL of culture medium, and part of the suspension was added to a new culture flask along with 5 mL of fresh culture medium.

### 2.2. Isolation and Activation of Human NK Cells

Peripheral blood NK cells (PB-NK) were isolated from healthy donors using the RosetteSep Human NK Cell Enrichment Cocktail (#15065, STEMCELL Technologies, Cambridge, MA, USA) according to the manufacturer’s protocol. After incubating peripheral blood with the reagent and centrifuging on a gradient of separation medium, CD56+ CD16+ NK cells were isolated. The purity of the isolated and expanded PB-NK cells was determined using flow cytometry analysis (Appendix A). Isolated NK cells from the donor were seeded at a density of 1 × 10^6^ per mL. NK cells were then washed from the remnants of the buffer and separation medium and activated by adding 300 U/mL IL-2 (SCI-STORE, Moscow, Russia) and 7 ng/mL IL-15 (SCI-STORE, Moscow, Russia) to the growth medium for 1 h.

### 2.3. Cell Viability Assay in Real-Time System

Cell proliferation and survival were analyzed using the iCELLigence RTCA (Real Time Cell Analyzer) system by measuring cell-to-electrode responses of the cells seeded in 8-well or 16-well E-plates with integrated microelectronic sensor arrays (ACEA Biosciences Inc., San Diego, CA, USA) as described previously [32].

E-plates containing 100 µL culture medium per well were equilibrated to 37 °C, and CI was set to zero under these conditions. Tumor (MCF7, MDA-MB-231, and SK-BR-3) and stromal (BrC4f and BN120f) cells were added to 500 µL of culture medium at a rate of 3 × 10^5^ cells/well for YT/PB-NK activity analysis and to 200 µL of culture medium at a rate of 2 × 10^3^ cells/well for the analysis of the effect of cytokines (IL-15/TGFβ) on cell viability. The culture medium was added to the control samples.

The YT cell line or PB-NK was added 48 h or 24 h, respectively, after target cell seeding in cancer/NK or stromal/NK cells at ratios of 1:1, 2:1, and 4:1. The CI was monitored in real time for 80 h after seeding. Cytokines IL-15 (50 ng/mL) or TGFβ (5, 10, 25 ng/mL) (SCI-STORE, Moscow, Russia) were added 48 h after seeding, and CI was monitored for 180 h.

### 2.4. Spheroids Formation

Homo- and heterotypic spheroids of BC were established using the liquid overlay technique in 96-well Nunclon™ (Waltham, MA, USA) Sphera™ (Chicago, IL, USA) U-shaped-bottom plates (#174925, Thermo Scientific, Waltham, MA, USA) [33]. Spheroids from stromal cells in multi-well agarose-coated plates. Homotypic models (3D) consisted of cancer cells (3De) or stromal cells (3Df), while heterotypic models (3D-2) consisted of cancer and stromal cells (1:4). Three independent experiments were performed (N = 3).

Cells were washed with 2 mL of 1× PBS and incubated at 37 °C for 3–5 min with 400 μL TrypLE™ (Waltham, MA, USA) solution. After detachment, the cells were washed with 1 mL of culture medium and counted using a LUNA-II^TM^ cell counter (Logos Biosystems, Anyang-si, Republic of Korea). After centrifugation (5 min, 330 rcf), the cells were suspended in 200 µL of DMEM/F12 containing 1× GlutaMAX™ Supplement (35050061, Gibco™, New York, NY, USA), 1 × Antibiotic-Antimycotic (15240062, Gibco™, New York, NY, USA), 20 ng/mL EGF (Epidermal Growth Factor; E9644, Sigma-Aldrich, Burlington, MA, USA), 20 ng/mL fibroblast growth factor basic (bFGF, PHG0261, Gibco™, New York, NY, USA), 5 µg/mL insulin (I9278, Sigma-Aldrich, Burlington, MA, USA), 2% B27 Plus Supplement (A35828010, Gibco™, Burlington, MA, USA), and 4% Albumin Bovine Serum Fraction V (BSA, 126593, Sigma-Aldrich, Burlington, MA, USA). This growth factor cocktail was composed as recommended by [24]. When working with immune cells, BSA was replaced with Panexin basic (P04-96900; PAN-Biotech, Aidenbach, Germany) to maintain NK cell viability. Cells in 3D culture media were seeded at a rate of 2500 cells/well. To construct heterotypic models, cancer cells were suspended with fibroblasts to reach a cancer/stromal cell ratio of 1:4, and the two cell types were seeded simultaneously.

Agarose hydrogel (2%, 50 µL) was added to each well of a 96-well culture plate (TPP, Trasadingen, Switzerland) and incubated at 37 °C for 1 h. Stromal cells were seeded in 100 µL of growth medium at a concentration of 2500 cells per well. The plates were incubated at 37 °C for an additional 72 h to allow the formation of 3D spheroids in culture.

### 2.5. Time-Lapse of Spheroid Formation Process

In order to analyze the process of spheroid formation, the CELENA X High Content Imaging System (Logos Biosystems, Anyang-si, Republic of Korea) was used. Cancer and stromal cells were seeded on a Nunclon™ Sphera™ U-shaped-bottom plate (#174925, Thermo Scientific, USA) as described previously, and the plate was then placed in a CELENA X incubation chamber, and time-lapse imaging of each well was conducted. The imaging time interval was 5 min on the first day of cultivation and 15 min on the following days. Three independent experiments were performed (N = 3). After the experiment was completed, the photographs were combined into a video file.

### 2.6. Fluorescence Microscopy


**Image Analysis NK Cell Penetration**


3D and 3D-2 were obtained as described in Section 2.4. To distinguish the different cell types within spheroids and evaluate NK cell penetration of spheroids, transduced cells expressing fluorescent proteins were used: tumor cells expressing mKate and fibroblasts expressing eGFP. BFP-expressing NK cells of the YT cell line in a 1:1 ratio (Effector:Target/E:T) or pre-stained Fluor 450 PB-NK cells in a 1:1 or 2:1 (E:T) ratio were added to spheroids on the 4th day of cultivation for SK-BR-3 3De models or on the 2nd day of cultivation for other models. The 3D and 3D-2 models were cultured with NK cells for 1 d (3Df spheroids) or 3 days (3De spheroids and 3D-2 models). Different techniques were used to visualize the penetration of NK cells into the spheroids.

The penetration of NK cell YT into 3Df spheroids was evaluated using a “flattened spheroids” preparation. For this purpose, the spheroids were transferred to polylysine glass and pressed with a cover glass. NK cell penetration was visualized using a Nikon Eclipse Ti-S series fluorescence inverted microscope (Nikon, Tokyo, Japan). Image analysis was carried out using the NIS-Elements software (Nikon Instruments Inc., version 5.30.05 64-bit, Melville, NY, USA).

The 3De and 3D-2 models with NK cell YT were visualized using a Nikon Eclipse Ti-S series fluorescence inverted microscope without flattening for the first 2 days of co-culture. On the 3rd day, spheroids were washed with PBS, transferred to a flat-bottomed plate (Eppendorf, Hamburg, Germany), and visualized using an LSM-510 Meta (Carl Zeiss, Oberkochen, Germany) confocal microscope. Three independent experiments were performed (N = 3). Images were analyzed using the Fiji software (ImageJ 2.16.0/1/54p, Java 1.8.0_442 (64-bit)).

### 2.7. Live/Dead Staining

To evaluate the effect of cytokines IL-15 (50 ng/mL) and TGFβ (5 ng/mL) (SCI-STORE, Russia) on PB-NK cytotoxicity, FDA/PI/Hoechst 33342 staining and fluorescence microscopy were performed on 3De, 3Df and 3D-2 models after 5 days of co-culture [34]. The staining was performed according to [35] with modifications. FDA is a non-fluorescent molecule that can be converted to fluorescein by intracellular esterases in viable cells with intact membranes. This means only live cells will show green fluorescence when stained with FDA. PI is a membrane-impermeant dye that enters cells only when the cell membrane is compromised (i.e., when the cell is dead or dying). Hoechst 33342 is a DNA-binding dye that has the capacity to stain all nuclei (both live and dead cells) due to its ability to passively diffuse into cells. In the event of PI or Hoechst being added first, there is the potential for damage to cells or alteration of membrane permeability, which may result in false-negative FDA staining (reduced fluorescein signal). Furthermore, PI has been observed to induce membrane disruption in certain instances, thereby impacting FDA hydrolysis [36]. Previously obtained spheroids were stained with 1 µg/mL fluorescein diacetate (FDA) (F1303, Thermo Fisher, USA) diluted in DMEM/F12 FBS-free medium for 45 min at 37 °C. After washing of FDA with 1× PBS, the spheroids were stained with 20 µg/mL propidium iodide (PI) (BD Biosciences, Franklin Lakes, NJ, USA) and 1:1000 Hoechst 33342 (Invitrogen, USA) diluted in PBS for 10 min at 37 °C. Three independent experiments were performed (N = 3). The results were visualized using a Nikon Eclipse Ti-S series fluorescence inverted microscope and image analysis was carried out using NIS-Elements software.

### 2.8. Calculation of the Spheroid Volumes

Spheroid volumes were assessed using light microscopy (Nikon Eclipse Ti-S) and Fiji (ImageJ) software on days 0 and 5 of cultivation, where day 0 is the spheroid volume before the addition of PB-NK cells and cytokines, and day 5 is the 5th day of co-cultivation. The spheroid sizes were calculated using the formula for the volume of a sphere V = 0.5 × L × W^2^, where L is defined as the diameter connecting the pair of farthest points on the spheroid contour, and W is the largest diameter perpendicular to L [37]. Three independent experiments were performed (N = 3).

### 2.9. Flow Cytometry

All analyses were performed using a FACS Canto II flow cytometer (BD Biosciences, Franklin Lakes, NJ, USA), and the data were analyzed using FACSDiva Software Version 6.1.3. (BD Biosciences). Cells were initially gated based on forward scatter versus side scatter to exclude small debris, and ten thousand events from this population were collected. The following antibodies were used for analysis: anti-MUC1-FITC from BD Pharmigen, NJ, USA. (#555925), anti-MICA/MICB-PE (#RT2204530) and anti-PD-L1-APC (#RT2568050) from Sony, anti-CD56-FITC (#MCA2693A488) from Stem Cell, anti-CD57 (#MCA1305GA), anti-CD62L-Alexa Fluor 700 (#MCA1076A700), anti-KIR-PE (#MCA2243PE), and secondary antibody Goat anti-Rabbit FITC (#5196-2404) from AbD Serotec.

The spheroids were collected, washed with 1× PBS twice, and dissociated with 500 µL Accutase^®^ (Capricorn, Ebsdorfergrund, Germany) for 10 min at 37 °C with 5% CO_2_ unless otherwise mentioned. To support the mechanical enzymatic dissociation of spheroids, the suspension was pipetted 10 times up and down to create shear forces. Accutase^®^ was neutralized by adding serum supplement media and centrifuged for 5 min at 1000 rpm. The supernatant was carefully removed after centrifugation. Finally, the cell suspension was placed on a pre-separation filter with a pore size of 70 µm (BD Biosciences, NJ, USA), which was filled with 800 µL PBS with 2% FBS, and centrifuged for 5 min at 1000 rpm. After supernatant aspiration, the cells were stained with a cocktail of antibodies and analyzed using flow cytometry. NucBlue™ Live ReadyProbes™ (Invitrogen™, USA) was added to the cell suspension just before detection using a flow cytometer. In addition to determining the cell number, the LUNA FL Cell Counter was used to quantify the number of single cells to monitor complete dissociation of the spheroids.

In order to facilitate the analysis of the PB-NK phenotype, the spheroids were harvested following co-culture with immune cells, and pre-purification with a 70 µm pore size filter (BD Biosciences, Franklin Lakes, NJ, USA) was performed to detach immune cells.

### 2.10. Statistical Analysis

Statistical analysis was performed on the results of three independent experiments. Spheroid formation time was analyzed using GraphPad Prism v.9.0 (GraphPad Software, San Diego, CA, USA), and Two-way Analysis of Variance (ANOVA) and Tukey’s HSD were performed. The results obtained are presented as the mean values of three repetitions and their standard errors.

The effects of PB-NK cells and cytokines (IL-2 with IL-15 or TGFβ) on spheroid volumes were analyzed using STATISTICA 10.0 software. Descriptive statistics were used to determine the group means and standard errors of the mean. Changes in spheroid volumes were analyzed using Repeated Measures ANOVA with gradations of the «experimental effect» factor (stimulation with TGFβ, IL-15, or no stimulation), followed by an assessment of intergroup differences using the Newman-Keuls post-hoc test. 

## 3. Results

### 3.1. Generation and Cultivation of 3D Heterotypic Breast Cancer Model

To generate a heterotypic spheroid breast cancer (BC) model, tumor cells such as MCF7, SK-BR-3, and MDA-MB-231, and patient-derived stromal cells such as cancer-associated fibroblasts (BrC4f) and normal fibroblasts (BN120f) were tested for their ability to form solid tumor spheroids. A schematic representation of the 3D heterotypic BC model generation is shown in the Graphical Abstract. Homotypic spheroids (containing only one cell type) from tumor or stromal cells (3D model) were used as a control for cellular interactions with NK cells. Homotypic stromal cellular spheroid formation was induced by seeding fibroblasts into 2% agarose-coated 96-well plates. The formation of homotypic and heterotypic spheroids (3D-2 model) was induced by seeding a mixture of tumor and/or stromal cells into a round-bottomed 96-well plate with ultra-low attachment [38]. During the aggregation process, cells fuse to form strings, which later round up to form spheroids (Figure 1). Cells assemble into loose cell aggregates and then undergo time-dependent formation of tight spheroids, i.e., the compaction stage. No correlation was observed between the spheroid formation time and the doubling time of the cells forming the spheroids (Figure 1d). We discovered that non-epithelial MDA-MB-231 and fibroblast cultures aggregated faster and formed denser structures than epithelialized spheroids (Figure 1a,c). It should be noted that the compaction of the model cell structure in spheroids formed from SK-BR-3 cells after the aggregation stage required at least 48 h. Spheroids formed by MCF7 and SK-BR-3 epithelial cells had a rounded shape and a well-defined inner edge, forming a dense capsule.

In the 3D-2 models, tumor cells adhered to large BN120f fibroblasts to form small aggregates, which then joined together to form spheroids, whose structure thickened during the co-culture period (Figure 1b). When tumor cells were co-cultured with BrC4f cells, uniform cell adhesion was observed, with fibroblasts in the center of the well surrounded by aggregating tumor cells. The time taken for spheroid formation was independent of the type of tumor cells within the models (Figure 1b). At the same time, the formation time of spheroids containing stromal cells varied depending on the cell line used for co-culturing. The addition of BN120f fibroblasts to the model led to faster spheroid formation, in contrast to BrC4f. Our previous study demonstrated the successful formation of heterotypic spheroids, along with the spatial rearrangement of stromal cells within the 3D structure [27,38]. It has been shown that the co-culture of tumor and stromal cells in the 3D-2 model leads to the formation of internal stromal layers among epithelial lobules. These layers were found to be similar cellular spheroids from tumor as controlcancer in vivo [27].

The formation of 3D and 3D-2 spheroids was used as a starting point for further experiments with the addition of the YT NK cell line after 4–5 days of spheroid maturation. NK cell activation can be triggered by growth factors, the addition of fetal bovine serum (FBS), human serum, plasma, or platelet lysates to the medium to maintain optimal proliferation of NK cells [39]. It is evident that FBS utilization does not yield optimal efficiency in generating 3D spheroid models, thereby exerting a discernible influence on cell metabolism and the process of spheroid formation [40]. Therefore, we used the Panexin basic serum substitute with specific components as an alternative to FBS. To investigate NK cell infiltration into a complex tumor model, the 3D-3 model was cultured in DMEM:F12 medium containing EGF, bFGF, insulin, B27, BSA, and 10%. Panexin was seeded in round-bottom 96-well plates with ultra-low attachment (Graphical Abstract).

The number of dead cells increased on day 10 in all model types, and a necrotic core formed in the 3D-2 spheroids (Appendix A). Therefore, in order to restrict the impact of cell death on the killing activity of NK cells, the 5th day of cultivation was chosen for further experiments with spheroids. On the 5th day, most cells in all types of spheroids were viable, and necrotic nuclei were not observed.

### 3.2. NK Cell Line Penetration in 3D Heterotypic Cellular BC Tumor Model

Initially, NK cell-mediated cancer cell killing was assessed in a 2D model using an xCELLigence real-time cell analyzer. The YT cell line was used as an “NK-like” cell line because it exhibits MHC-unrestricted cytotoxicity, does not respond to IL-2, lacks expression of CD3, and has no rearranged TCR-β genes [41]. Given that the NK-like cell line YT is not IL-2-dependent and lacks several other hallmarks of NK cells, such as CD16 expression, we also examined primary NK cells isolated from the peripheral blood of healthy donors (PB-NK). YT cells were added to the target cells at a 1:1 ratio (E:T). In order to illustrate the cytotoxic potential of PB-NK cells, the effect was examined by adding the cells to target cells at varying ratios.

YT and PB-NK cells demonstrated differential toxicity against in 2D tumor cells and fibroblasts (Appendix A). Mesenchymal-like MDA-MB-231 cells and all fibroblast cultures were lysed by YT cells, in contrast to epithelial tumor cells (MCF7, SK-BR-3), which exhibited resistance to YT-mediated lysis. Conversely, PB-NK cells exhibited marked efficacy in eliminating epithelial cancer cells MCF7, SK-BR-3, and MDA-MB-231, with this efficacy being both dose- and time-dependent. PB-NK cells also suppressed the proliferation of patient-derived fibroblasts, with BrC4f and BN120f being particularly sensitive, even at low-to-average E:T ratios.

The infiltration of NK cells is a prerequisite for tumor spheroid destruction [42]. mKate2-expressing MCF7, SK-BR-3, and MDA-MB231 cells were used as tumor components of the 3D spheroids. In order to form 3D-2 spheroids, eGFP-expressing and patient-derived normal BN120f fibroblasts were used as an “extra” stromal component [27]. BFP-expressing NK cell line YT cells were added at a 1:1 E:T ratio after the formation of 3D and 3D-2 spheroids. The degree of NK cell cytotoxicity was monitored based on the number of target cells exhibiting persistent fluorescence. Subsequently, NK cells quickly attached themselves to the 3De structure and formed clusters on its outer ring. In this model, NK cells showed no cytotoxic activity against tumor spheroids after three days of co-culture, which was confirmed by the preservation of spheroid structural integrity (Figure 2). MUC1 is generally overexpressed in cancer cells, which could also be associated with protection against NK cells, regardless of the expression levels of NK-associated receptors [43]. The formation of a mucin capsule around the spheroid from MCF7 cells may be responsible for these results, limiting the interaction of NK cells with tumor cells [44]. 

CAFs can promote NK cell migration through the tumor matrix, potentially leading to increased infiltration into the spheroids [6]. NK cell killer activity in 3D-2 models of MCF7 BC cells revealed the close proximity of NK cells to stromal cells, which formed an inner core for the outer tumor layer. In the case of 3D-2 spheroids of SK-BR-3 cells, NK cell cytotoxic activity was directed against the surface layer cells of the spheroids (Appendix A). Immune cells significantly disrupted the architecture of 3D-2 spheroids of MDA-MB-231. It appears that NK cells initiate an attack on the spheroid from the surface layer, gradually penetrating the interior to exert a cytotoxic effect on the cells layer by layer (Appendix A). Consequently, on the 2nd day of co-culture, only a small spherical fragment of the spheroid remained after NK cell addition (Figure 2b). In models containing CAF BrC4f, the stromal cells were located centrally within the spheroid.

The destruction of the spheroid structure can be regarded as an indication of the cytotoxic activity of NK cells against 3D models. The infiltration of immune cells was only observed in heterotypic spheroids from MCF7 and MDA-MB-231 cells, which is likely related to the activity of fibroblasts. NK cells were suggested to interact most actively with fibroblasts and MDA-MB-231 mesenchymal-like cells. However, the reason for the differences in the interaction with BrC4f CAFs in the models containing MCF7 and MDA-MB-231 cells requires further investigation (Figure 3).

Next, we analyzed natural killer cell interactions with 3Df stromal spheroids. These cells rapidly bound to the outer periphery of the spheroids and formed clusters (Figure 4). However, these NK cell clusters did not exhibit killing activity. Thus, it can be concluded that CAF within the tumor microenvironment are involved in the regulation of natural killer cell activity.

In heterotypic spheroids, PB-NK cells (at high E:T ratios of 1:1 and 2:1) induced a significant loss of mKate2 (tumor cells) and GFP (CAFs) fluorescence by days 1 and 3, respectively, demonstrating potent toxicity compared to YT cells (Figure 2 and Appendix A). In light of these findings, subsequent experiments were devised to evaluate the impact of mean PB-NK E:T ratios in conjunction with cytokines (IL-15/TGFβ) on immune cell infiltration within 3D models. It is hypothesized that these experiments could open up the therapeutic potential of allogeneic NK cells in anti-cancer treatment regimens.

### 3.3. Cancer-Associated Fibroblasts Regulated MIC A/B, MUC1 and PD-L1 Expression on Tumor Cells in Spheroids

NK cell functions are regulated by the balance of activating and inhibiting signals triggered by membrane receptors expressed by NK cells and their corresponding ligands on the target cells [45]. The major histocompatibility complex-class I chain-related proteins A and B (MIC A/B) are usually upregulated in cancer cells because cellular stress and MIC A/B shedding by cancer cells can cause an escape from NKG2D recognition, favoring the emergence of cancers [46]. PD-L1 plays a critical role in suppressing NK cell function by binding to the PD1 receptor on NK cells [47]. Mucin (MUC1) can also inhibit target cell lysis by NK cells [48]. To explain the distinct sensitivity of tumor and stromal cells to NK-YT cytotoxicity, the expression levels of MIC A/B, PD-L1, and MUC1 were investigated (Figure 5).

All cell models showed increased MIC A/B expression when grown in three dimensions, with an exception of SK-BR-3. When we cultivated the tumor cells with BrC4f, the MIC A/B expression on the surface of the 3D-2 spheroids increased dramatically (Figure 5a). A considerable increase in the expression of MIC A/B and MUC1 was observed in MCF7 cells in the presence of CAF BrC4f in 3D-2 without high killer activity of YT (Figure 2a and Figure 5c). It was also demonstrated that MUC1 overexpression in cancer cells is associated with a heightened capacity for defense against natural killer cells, and that this process is regulated independently of MIC A/B [43].

Estrogen receptor-positive (ER+) breast cancers as 3De models from MCF7 are often described as «immunologically cold», meaning they have a low immune response [49]. The results showed an increase in PD-L1 expression when the samples were cultured under 3D conditions, with the exception of the MCF7 cell line (Figure 5b). Early-stage triple-negative BC 3De models derived from MDA-MB-231 cells have been shown to exhibit either an “immunologically hot” or “immunologically cold” phenotype when co-cultured with different cell types. Understanding the mechanisms underlying this phenotypic variability may provide valuable insights into guiding treatment strategies [50]. In the heterotypic spheroids from MDA-MB-231 with stromal cells, a decrease in PD-L1 levels was observed, which in turn resulted in increased sensitivity to YT cells (Figure 2b). Analysis revealed that co-culturing heterotypic spheroids derived from other tumor cell types with BrC4f cells leads to upregulation of PD-L1 expression. The MFI values for the 2D cellular models are presented in the Appendix A.

The presence of CAFs within the model induced alterations in the expression levels of MICA/B and PD-L1 by breast cancer tumor cells within the 3D-2 model. This observation underscores the pivotal role of CAFs in modulating the response to immunotherapy. 

### 3.4. The Combination of Cytokines and NK Cells Increases Tumor Cell Killing

Different types of cytokines can be used to activate and expand NK cells for cellular therapy. IL-15 and IL-2 were added to the resulting immune cells when PB-NKs were isolated from PB to maintain their viability. Using xCELLigence real-time analysis, we found that TGFβ suppressed MCF7 and SK-BR-3 proliferation at all tested concentrations (5–25 ng/mL), while enhancing MDA-MB-231 growth, particularly at near-physiological concentrations. IL-15 inhibited only MDA-MB-231 cells and had no effect on the other tumor cells (Appendix A). In stromal cells, TGFβ and IL-15 suppressed BN120f fibroblast proliferation (strongest at 25 ng/mL TGFβ) but boosted CAF BrC4f growth, with peak TGFβ effects at physiological doses (5–10 ng/mL) (Appendix A). 

In the subsequent experiment, the potential to modulate the immune response in the tumor model was investigated. The dynamics of spheroid destruction were monitored for five days, with differential staining of all spheroid cells with FDA+ (live, green signal) and PI+/Hoechst 33342+ (dead, red/blue signals) for the different stages of 3D-structure formation. In addition, the dynamics of spheroid volume changes were used as an assessment tool. PB-NK cells were added to the spheroids at an E: T ratio of 1:2. Furthermore, both NK cells and cytokines were added to 3D-3 concurrently. Cytokines IL-15 (50 ng/mL) or TGFβ1 (5 ng/mL) were added to the heterotypic spheroids at physiological concentrations.

In the case of 3De derived from MCF7, a substantial number of viable cells (FDA+) were observed in both the control and IL-15 or TGFβ-treated groups (Figure 6a). However, 3De models derived from SK-BR-3 cells revealed the absence of PB-NK cell activity, both in the absence and presence of IL-15 supplementation. Conversely, the addition of TGFβ to the spheroids resulted in the absence of viable cells (PI+/Hoechst 33342+), corroborating the data from xCelligence (Figure 6a and Appendix A). The addition of PB-NK cells to MDA-MB-231 models resulted in the demise of all types of spheroids. Experiments conducted with MDA-MB-231 spheroids demonstrated that IL-15 stimulation induced PB-NK cell activity. The enhanced presence of viable cells within the spheroids compared to the control group after the incorporation of TGFβ may be ascribed to either the suppression of natural killer cell activity or a direct effect of TGFβ on the cells within the models (Figure 6 and Appendix A). The study revealed no statistically significant variations in the volumes of 3D models derived from MCF7 and SK-BR-3 BC cells, irrespective of the stimulation. Conversely, PB-NK and cytokine co-exposure to the spheroids led to a decrease in the volumes of 3D models derived from MDA-MB-231 cells, in contrast to the control group without the addition of PB-NK (Appendix A).

In 3D-2 spheroids derived from MCF7 cells, a decrease in the number of live (FDA+) cells was observed when IL-15 was added compared to the control (Figure 6b,c). Interestingly, only a necrotic core (PI+/Hoechst 33342+), most likely consisting of stromal cells, remained in 3D-2. The addition of TGFβ resulted in the probable inhibition of PB-NK cell activity for 3De from MCF7 and models containing CAFs, but not for models containing normal fibroblasts. For 3D-2 models from SK-BR-3 cells, upon the addition of both IL-15 and TGFβ, a necrotic core was observed within the spheroid, irrespective of the fibroblast type. However, a small number of viable cells were still present in the spheroids with BrC4f (Figure 6b,c). The addition of PB-NK cells to the 3D-2 models from MDA-MB-231 cells ensured the viability of cells within the CAF, thus obviating the necessity for any further stimulation. The results showed that TGFβ promoted the proliferation of MDA-MB-231 and BrC4f cells and inhibited the killer activity of PB-NK cells (Appendix A). A higher number of viable cells was observed in the spheroids from MDA-MB-231 cells compared to the 3D-spheroid and 3D-2-spheroid containing normal fibroblasts (Figure 6b,c). The investigation revealed that co-exposure to PB-NK and cytokines exerted no substantial influence on the volume of 3D-2 from MCF7 with all types of fibroblasts. In the present experiment, the treatment of 3D-2 from SK-BR-3 with fibroblasts significantly decreased the spheroid volume in both the control and experimental groups. The combination effect of PB-NK and cytokines on 3D-2 models from MDA-MB-231 with CAF resulted in a significant increase in spheroid volume (Appendix A). This may be due to the increased friability of the spheroids as a consequence of their destruction by the combination of PB-NK and cytokines.

We also conducted a similar experiment in 3D cultures of stromal cells. The 3D CAFs BrC4f exhibited heightened sensitivity to IL-15 or TGFβ1-activated PB-NK cells (Figure 7). Stimulation of PB-NK cells with TGFβ resulted in FDA-positive cells (live cells) and fine spheroids from normal fibroblast cell culture BN120f. The study revealed statistically significant variations in the volumes of the 3D models (Appendix A).

In summary, the present study demonstrated the activation of PB-NK cell cytotoxicity after treatment with IL-15, as observed in most models. In contrast, no such inhibition of PB-NK cell activity was detected in the presence of TGFβ due to the direct effect of this cytokine on the cell population within the spheroids, resulting in their death across all models except those containing MDA-MB-231. The findings from the 2D cultures were consistent with those from the 3D and 3D-2 models.

### 3.5. Evaluation of PB-NK Phenotype Changes Within a 3D Model upon Exposure to IL-15 or TGFβ1

During differentiation and activation, the phenotype of NK cells undergoes modification, resulting in the formation of distinct subpopulations that possess divergent functional characteristics. Following co-culture of PB-NK cells and spheroids of MCF7 with stromal cells/cytokines, including IL-15 (50 ng/mL) or TGFβ1 (5 ng/mL), the infiltrating cells and those remaining in the medium were mechanically separated, after which the phenotype of PB-NK cells was analyzed.

In this study, isolated PB-NK cells were defined as a CD56-dim population of immune cells (Appendix A). The addition of PB-NK cells to 3D-2 resulted in a minimal (2–3%) augmentation of the CD56-bright NK cell population, irrespective of the fibroblast type present in the model, compared to 3D from tumor cells (Figure 8a). In addition, a slight decrease in the total percentage of CD56+ cells was observed compared to the PB-NK cell population cultured alone without target cells (blue histogram). As illustrated by the green histogram, the introduction of IL-15 did not impact the alterations in the number of CD56+ cells except for those observed in the model comprising normal fibroblasts. The addition of TGFβ (dark green histogram) decreased the total number of CD56+ cells and CD56-dim NK cells, which was most pronounced in 3D-2 with BN120f (Figure 8a). The addition of PB-NK cells to stromal spheroids from CAFs BrC4f resulted in a marked increase in CD56-bright NK cells (Figure 8b). In addition, the data demonstrated a decrease in the percentage of CD56-dim NK cells among all fibroblasts. IL-15 stimulation (light green histogram) increased the number of CD56-bright NK cells in co-cultured fibroblasts derived from the CAFs line, while concurrently reducing the percentage of CD56-dim cells. The data show that the number of CD56+ NK cells in CAFs increased compared to that in unstimulated PB-NK cells (orange and blue histograms). As illustrated by the dark green histogram, TGFβ1 stimulation inhibited CD56-dim NK cells. In addition to this observation, it has also been demonstrated that TGFβ1 stimulation results in an increase in the percentage of CD56-bright NK cells in the 3D CAF model (Figure 8b).

To assess the initial NK cell phenotype, the MFI of CD57 (Figure 9), CD62L (Figure 10), and KIR (Figure 11) on NK cells at baseline was evaluated considering the above-defined groups. CD57, which is induced in the CD56-dim subpopulation, has been identified as a marker of terminal differentiation with high cytolytic activity [42]. Analysis of the MFI of CD57 (Figure 8) indicated that stimulation with IL-15 (light green histogram) or TGFβ1 (dark green histogram) did not change the percentage of the cell population. Interactions between immune cells and tumor cells in 3D have been shown to result in a decrease in the MFI of CD57 on PB-NK cells. In the 3De model, a discrepancy was apparent in the MFI of CD57 values when compared with the control group, which was not subjected to additional stimulation (orange histogram). The present study demonstrated that stimulation with TGFβ1 resulted in a significant increase in the MFI of CD57 (dark green histogram) in 3De of MCF7 cells. For 3D-2, there was a non-significant increase in the MFI of CD57 for the models containing CAF (Figure 9a). In the case of 3Df, stimulation with IL-15 or TGFβ1 resulted in a significant increase in the MFI of CD57 in CAF-containing spheroids. Conversely, TGFβ1 stimulation has been found to decrease the MFI of CD57 in normal fibroblasts in the 3Df model (Figure 9b).

The cell adhesion molecule CD62L may also be indicative of the degree of differentiation of mature circulating NK cells. The present study demonstrated that stimulation of PB-NK cells and cells in spheroids via TGFβ or IL-15 resulted in alterations in the percentage of CD62L+ PB-NK cell population and decreased MFI of CD62L in the 3De and 3D-2 models with CAFs compared to non-stimulated PB-NK cells (orange histogram) (Figure 9a). The 3D-2 model with normal fibroblasts showed a decline in the percentage of CD62L + PB-NK cells, with no alteration in MFI values. In the case of 3Df, stimulation with IL-15 or TGFβ resulted in a significant increase in the MFI of CD62L for CAF, while concurrently resulting in a decrease in the MFI of CD62L for normal fibroblasts (Figure 10b).

The maturation of NK cells can be traced by the appearance of one or more types of KIR receptors on the cell surface (Figure 11). An analogous tendency with CD62L in the 3De model from MCF7 was observed during the evaluation of alterations in KIR receptor activation. This encompassed the populations and MFI of KIR values in comparison to controls that lacked additional stimulation (Figure 11a). The incorporation of normal fibroblasts into 3D-2 with cytokine stimulation did not affect KIR expression on the surface of NK cells. This outcome contrasts with that observed following the addition of CAFs, which resulted in an increase in MFI. As demonstrated in the case of 3Df, the application of IL-15 or TGFβ resulted in a substantial augmentation of the MFI of KIR within the model comprising CAFs (Figure 11b).

It could be concluded that PB-NK stimulation with IL-15 or TGFβ1 has an effect on the activation of CD57, CD62L and KIR on PB-NK cells. This effect is contingent upon the type of stromal cells in heterotypic cellular 3D tumor models.

## 4. Discussion

In order to understand how each component of the TME affects tumor behavior and to generate ideas for treating tumors, it is essential to understand the physiological role of each element of the TME. In this study, we present a 3D heterotypic spheroid model of breast cancer consisting of tumor, stromal, and immune cells with an environment that mimics cellular interactions. Our previous study demonstrated that co-culturing tumor and stromal cells was associated with a higher propensity for the formation of rounded and structured spheroids compared to those formed from tumor cells alone. The findings of this study demonstrated a correlation between the type of tumor cells utilized in the SK-BR-3 ≥ MCF-7 > MDA-MB-231 series and a decline in both tumor size and spheroid heterogeneity [27]. The investigation revealed that fibroblast cells constituted the inner core of the spheroid, while epithelial tumor cells comprised the outer layer of 3D-2 structures [51,52].

There are three stages in the process of spheroid formation: cell aggregation due to the binding of integrin and extracellular matrix (ECM) components, a delay associated with the expression and accumulation of E-cadherin, and compaction correlated with the interaction of E-cadherin molecules with each other [53]. In our study, we showed that spheroid formation time was independent of tumor cell type in the models, and there was no correlation between spheroid formation time and doubling time of cells forming the spheroids (Figure 1). However, when normal fibroblasts were added to the model, spheroids formed more rapidly than when CAFs were added, resulting in a statistically significant difference in the formation time of models containing different types of stromal cells. We have previously shown the ability of stromal cells to aggregate tumor cells into heterotypic spheroids of BC [27]. The stroma will become increasingly appreciated in the initiation or formation of breast carcinomas [54]. The mechanism by which fibroblasts support spheroid formation is not fully understood; however, they are known to produce extracellular matrix (ECM) components that attract cancer cells and use their actin cytoskeleton to contract and organize cell populations [55,56]. The stroma, which supports the epithelial tissue, primarily consists of ECM components [57]. There is strong evidence that changes in the stroma drive the initiation of epithelial tumorigenesis. In a mouse experiment, Maffini et al. showed that a carcinogen can trigger the growth of epithelial tumors by altering the adjacent stroma [58]. In addition, variations in the velocity of spheroid formation may be attributable to dis-parities in the secretion of ECM constituents by normal fibroblasts and CAFs, which promote spheroid cell adhesion (Figure 1).

NK cells play a crucial role in triggering antitumor immune response [4]. Although NK cells hold promise for cancer therapy, NK cell infiltration into tumors and their phenotype and function within the tumor have not yet been extensively investigated, partly due to a lack of appropriate model systems [59]. Interestingly, when we added YT cells to 3D heterotypic cellular tumor models, YT penetrated the spheroid and remained targeted to the fibroblast-rich core, binding to fibroblast cells (Figure 2, right panel). Moreover, tumor cells in such 3D-2 spheroids from MDA-MB-231 cells restored the sensitivity to the killer activity of NK cells. However, spheroids from different types of breast fibroblasts demonstrated resistance to YT (Figure 3). No sensitivity to YT was observed in the SK-BR-3 and MCF7 cell lines in the 2D models (Appendix A). In the 3D-2 models containing MCF7, NK cells were localized next to the stromal cells, acting as an internal framework for the outer tumor layer, which may provide an indirect indication of NK cell infiltration and killer cell activity in the models (Figure 2 and Figure 3). As demonstrated in Figure 2, the results obtained from 3D and 3D-2 spheroids derived from SK-BR-3 cells are consistent with the viability assay of spheroids from SK-BR-3 cells (Appendix A). This finding indicates that the cytotoxic activity of NK cells in this case was directed against the cells on the surface layer of the spheroids. In this case, it can be assumed that 3 days of co-culture is not enough to significantly disrupt the structure of the spheroids, which have large dimensions. As demonstrated in previous studies, the spheroids derived from SK-BR-3 cells exhibited significantly larger dimensions (up to 1 μm) than all other models contemplated in the present study [27]. A plethora of preclinical and clinical data have been published, which suggest that the efficacy of NK cells in combating solid malignancies is impeded by several factors. These include difficulties in tumor infiltration and persistence/activation within the tumor microenvironment. The metabolic features of the tumor microenvironment, including hypoxia and elevated levels of adenosine, reactive oxygen species, and prostaglandins, have been demonstrated to exert a detrimental effect on the activity of natural killer cells [60]. As demonstrated in Appendix A, NK cells did not penetrate the spheroid and instead exhibited killer activity layer by layer, initially attacking cells at the periphery of the model and gradually moving towards the center, thereby confirming the difficulties previously described in preclinical and clinical studies [61]. The cell lysis effect of NK-like YT cells was different from that of PB-NK cells: we identified that YT inhibited the growth of non-epithelial MDA-MB-231 breast cancer cells and patient-derived fibroblasts, such as BrC4f and BN120f, in the 2D model (Appendix A). However, YT cells did not inhibit the growth of cells in the 3D model and remained only on the external layer (Figure 2b). This effect may be related to the ability of stromal cells to produce stress-inducible ligands during prolonged in vitro cultivation and the KIR mismatch between YT cells and fibroblasts. This off-target behavior can also be explained by the graft-versus-host effect, where immune cells from one donor can respond to the normal cells of the recipient with a mismatched haplotype [49]. The sensitivity of tumor and stromal cells to PB-NK changed in the following order: BN120f ≥ BrC4f > MDA-MB-231 ≥ MCF7 > SK-BR-3 (Appendix A). PB-NK cells exhibited cytotoxic activity against BN120f and BrC4f fibroblasts; however, the effect was independent of the ratio of added NK cells to target cells. It can be assumed that the observed differences in the effects of PB-NK cells on tumor cells are due to the presence of MHC-I ligands on the targeted cells. It has been shown that the MHC-I protein level is 18% of that of MDA-MB-231 (where the MHC-I level of the MDA-MB-231 cell line is taken as 100%) for MCF7 cells and 0% for SK-BR-3 cells [62]. The absence of MHC-I in these cell lines shifts the receptor balance of PB-NK cells towards activation and manifestation of cytotoxic functions.

Human tumors frequently express the MICA and MICB ligands of the activating NKG2D receptor; however, proteolytic shedding of MICA/B represents an important immune evasion mechanism in many human cancers [63]. A comparison of the differences in expression levels between CAFs and normal fibroblasts cultured under 3D conditions revealed that CAFs exhibited increased levels of MICA/B and PDL1 (Figure 5 and Appendix A). Ziani and coworkers demonstrated that melanoma-associated CAFs decrease the sensitivity of melanoma tumor cells to NK cell-mediated killing through the secretion of MMPs, which cleave MICA and MICB at the surface of the tumor cells and consequently decrease both NKG2D-dependent cytotoxic activity of NK and their secretion of IFNγ [45]. Most likely, one of the key players in the process of NK cell activation is the activating receptor NKG2D, which initiates downstream signaling through the PI3K/PLCgamma/Jak2 pathway [64]. In our study, we found that the MFI of MIC A/B increased in the 3D-2 model when cultured with CAFs and decreased the activity of the immune cell line YT (Figure 2 and Figure 5a). In addition, spheroids from the MCF7 cell line were characterized by increased MUC1 expression, which may also protect the tumor from immune cell infiltration (Figure 5c). Dhandapani et al. demonstrated that the in vitro 3D spheroid model preserves the TME of the “immune hot” (MDA-MB-231) and “immune cold” (MCF7) breast cancer subtypes [65]. By secreting various chemokines, cytokines and other effector molecules, CAFs directly inhibit immune cell-mediated antitumor immunity [66]. Morimoto and colleagues demonstrated that MUC1-C represses the expression of MICA and MICB ligands that activate NKG2D [43]. Maeda et al. showed that in triple-negative BC, MUC1-C upregulation recruits MYC and NF-κB p65 to the PD-L1 promoter, enhancing PD-L1 expression and leading to immune escape and reduced patient survival [67]. We found that the 3D-2 model of MDA-MB-231 with CAFs was characterized by a decrease in MFI of PD-L1 levels and increased NK sensitivity (Figure 2b and Figure 5b). Fibroblasts have been shown to promote the differentiation of CD56-bright NK cells into mature effector CD56-dim cells with increased cytotoxicity [68]. Moreover, CAFs can secrete factors that contribute to an immunosuppressive microenvironment (e.g., TGFβ), which suppresses immune cell activity [10]. PD-L1 expression is also a key factor in maintaining the invasive phenotype of fibroblasts [69]. In 3Df BC, as a fibrosis model of CAFs, the MFI of PD-L1 increased. Elevated PD-L1 levels could be directly or indirectly involved in the fibrotic response by regulating the activation of fibroblasts and their immunomodulatory capacity. This highlights the importance of studying tumor cell-microenvironment crosstalk.

Cytokines are pivotal for the maturation, activation, and survival of NK cells [70]. Due to resistant tumor subpopulations and soluble tumor microenvironment factors such as TGFβ1, the use of cytokines as agents to enhance the immune response against malignancies may not always be effective. Herter and colleagues studied the infiltration of immune cells in benign and malignant cell spheroids and showed that stimulation of human peripheral blood mononuclear cells (PBMCs) by IL-2 was necessary for the infiltration of PBMCs into the spheroids [29]. In our work, we used IL-2 (300 units/mL) and IL-15 (7 ng/mL) for 1 h to stimulate NK cell activation/expansion, and then used IL-15 (50 ng/mL) or TGFβ (5 ng/mL) for the ex-periment with 3D models of BC. Activated PB-NK cell cytotoxicity induced by IL-15 was observed in the majority of models. Inhibition of PB-NK cell cytotoxicity after treatment with TGFβ was not detected due to the direct effect of this cytokine on the cells in the spheroids, leading to their death in all cases except for the models with MDA-MB-231. The data from 2D cultures is in agreement with the results from the 3D models (Figure 5 and Appendix A). Rather than suppressing tumor growth, TGFβ1 may interfere with the activity of NK cells by activating or inhibiting specific mechanisms that disrupt the expression of activating receptors and alter cellular metabolism [53]. TGFβ1 blocks NK cell activation and inhibits its cytotoxic potential by directly or indirectly blocking the expression of the C-type lectin receptor, NKG2D [71]. TGFβ1 blocks NK cell activation and inhibits its cytotoxic potential by directly or indirectly blocking the expression of the C-type lectin receptor NKG2D [72]. In our experiments, only for MDA-MB-231, which mimics triple-negative BC, we found that the addition of TGFβ1 to the heterotypic cellular tumor model suppressed PB-NK cell activity and increased tumor cell proliferation (Figure 6 and Appendix A). Foltz et al. showed that activation of NK cells by IL-2 and TGFβ1 leads to a pro-inflammatory phenotype and induces hypersecretion of IFNγ and TNFα by NK cells in response to tumor cells [73]. This could potentially explain the lack of inhibition of PB-NK cell activity in MCF7, SK-BR-3, and stromal cells (Figure 6 and Figure 7). The immunosuppressive role of TGFβ1 on NK cells has been described by Chung et al. in 2024 [16]. The mechanism of CD4+ T cell suppression by TGFβ1/IL-15–induced CD103 + CD56+ NK-like cells included a significant reduction in CD25 expression on the surface of T cells. In contrast, TGFβ1 could have an immunosuppressive effect on NK cells due to its ability to induce the conversion of anti-tumor NK cells into pro-tumor ILC1s (type 1 lymphoid cells) [74]. A recent study on downstream TGF-β signaling identified TAK1-mediated activation of p38 MAPK as the critical pathway driving conversion. IL-15 enhances TGF-β-mediated conversion through Ras:RAC1 signaling as well as via the activation of MEK/ERK [75]. These effects support our observations regarding the complex role of TGFβ1 and highlight possible directions for comprehensive further studies.

We propose that the development of an adaptive NK cell pool may be associated with changes in the characteristics of highly differentiated cells within the tumor microenvironment (TME). CD56-dim NK cells are fully mature NK cells that account for approximately 90–95% of PB-NKs, and they kill target cells by means of antibody-dependent cell-mediated cytotoxicity. Upon exposure to IL-2 and/or IL-15, CD56-bright NK cells differentiate into CD56-dim cells [68]. NK cell-mediated tumor cell death is inhibited by CAFs. The addition of IL-15 to the experimental models resulted in a marked increase in CD56+ cells, while the addition of TGFβ1 led to a decrease in the total number of CD56+ cells and CD56-dim NK cells. The combination of IL2 with IL-15 or TGFβ1 resulted in a substantial increase in CD56-bright NK cells within co-cultured CAFs, accompanied by a concurrent decline in CD56-dim NK cells. This phenomenon led to an increase in CD56-NK cells in the presence of normal fibroblasts. Combination stimulation of PB-NK with IL-2, IL-15, or TGFβ1 affects the activation of CD57, CD62L, and KIR PB-NK cells and, depending on the type of stromal cells within 3D-2, increases (CAF) or decreases it (normal fibroblasts). (Figure 9, Figure 10 and Figure 11 and S11). The addition of IL-2 promotes the functional maturation of CD56-bright NK cells in lymph nodes and increases the expression of KIR, CD16, NCR, and perforin [76]. Some studies have shown that IL-2 and IL-15 activate the expression of NKG2D receptors, NCR family members, and KIR family members [77]. This may contribute to the recognition and killing of the target cells. CD56-dim cells with an intermediate signature (CD94/NKG2A+, CD62L+, or CD57) are known to combine the ability to produce IFN-γ [78]. The ability of the CD56-bright subpopulation of NK cells to respond to cytokine stimulation was strongly correlated with CD94, CD62L, and CD57 expression. This reflects the higher cytokine receptor expression and STAT4 activation. In contrast, the capacity of CD56-dim subpopulations to eliminate tumor cells and generate IFN-γ in response to actR was predominantly associated with the expression of intrinsic inhibitory MHC-I-binding receptors in KIR family members [79]. TGFβ1 alters the metabolic profile of NK cells, limits their cytotoxicity, and mediates the transformation of NK cells into intraepithelial ILC1-like NK cells [80] with enhanced effector functions [81]. CD57, which is induced in the CD56-dim sub-populations, is a marker of terminal differentiation with a high level of cytolytic activity [82]. Cultivation of CD56-dim and CD56-bright with IL-15 and stromal cell lines has resulted in proliferation, KIR expression, cytotoxicity, and cytokine production [83]. 

In conclusion, the heterotypic cellular 3D tumor model of breast cancer, composed of tumor cells, fibroblasts, and immune cells, presented here allows for the investigation of immune cell infiltration and modification of immune cell properties in the TME in vitro. This allows a comprehensive in vitro evaluation of immune cell infiltration and target cell clearance after treatment, which cannot be achieved using 2D systems. However, little is known about the influence of CAFs on NK cells in the context of carcinomas. Targeting fibroblasts could help “warm up” the tumor microenvironment, including activating NK cells, opening possible avenues for new biomarkers and strategies for immune-based therapies.

## 5. Conclusions

The development of useful 3D cancer models for drug testing is currently in an exploratory phase, aiming to better capture the complexity and dynamics of the underlying interactions within the tumor microenvironment (TME) in a more physiological manner [55]. The characterization of the functional state of NK cells interacting with different tumor microenvironment cells may vary at different cancer stages and may help identify new therapeutic strategies. Several data describing NK cells in tumors may also be markers of the critical impact of NK cells on recurrence in patients with early-stage breast cancer [84,85]. Therefore, the question was raised whether NK cells can infiltrate the innermost layers of the tumor, lacking stromal components in patients with progressive-stage breast cancer, and whether there are non-specific cell-to-cell interactions between NK cells and different types of fibroblasts in the TME. Our model is designed to better represent and characterize the mutual influencing factors of fibroblasts and tumor cells. Fibroblast-supplemented heterotypic spheroids are a valuable tool for TME interactions and new drug discovery.

CAFs represent a promising target in oncology or immuno-oncology, potentially augmenting current tumor therapies or enhancing the immune response. Fibroblasts, or their presence in 3D cancer models, provide an opportunity to test cancer drugs that target the fibroblasts, such as PDGFR-β and FAP-α. The aim of this study was to characterize the influence of fibroblasts on tumor cells and evaluate the activity of natural killer (NK) cells in the tumor microenvironment (TME) to better understand their effects on tumor behavior. 3Df mimics fibrotic tissue with enhanced rigidity due to the deposition of ECM proteins, which can make immune cell infiltration more difficult. These findings highlight the critical differences between adherent and spheroid stromal cell interactions with human immune cells, which have significant translational consequences. Some immunotherapies are based on immunomodulation and the addition of exogenous molecules, such as cytokines (e.g., IL-2) or antibodies targeting tumor-immune cell interactions (e.g., PD-1/PD-L1 or HER-2). Currently, resistance to immunotherapy is associated with a cold immunological tumor microenvironment and recirculation of target receptors. Interestingly, according to our observations, in the presence of fibroblasts, tumor cells increased PD-L1 expression level, which indicates additional mechanisms of drug resistance associated with fibroblasts.

Overall, this study establishes a foundation for incorporating the tumor environment into considerations regarding tumor behavior and response to therapeutics. Therefore, it is essential to further develop mimicking models.

## Figures and Tables

**Figure 1 cells-14-01039-f001:**
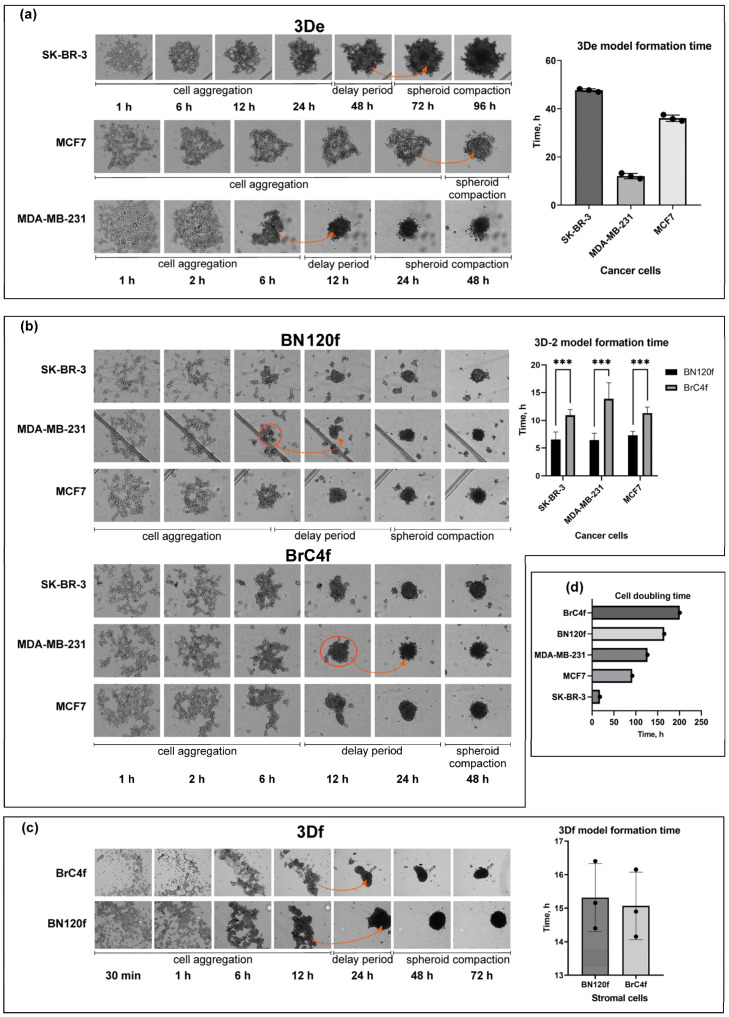
Time-lapse transmitted light imaging formation of 3D and 3D-2 spheroids. Data sets out the growth, step and formation time of the spheroid from (**a**) tumor cells; (**b**) mixture of cells; (**c**) stromal cells of BC; (**d**) comparison of cell growth, population doubling time. *** *p* < 0.001 determined by unpaired two-tailed Student’s *t* test.

**Figure 2 cells-14-01039-f002:**
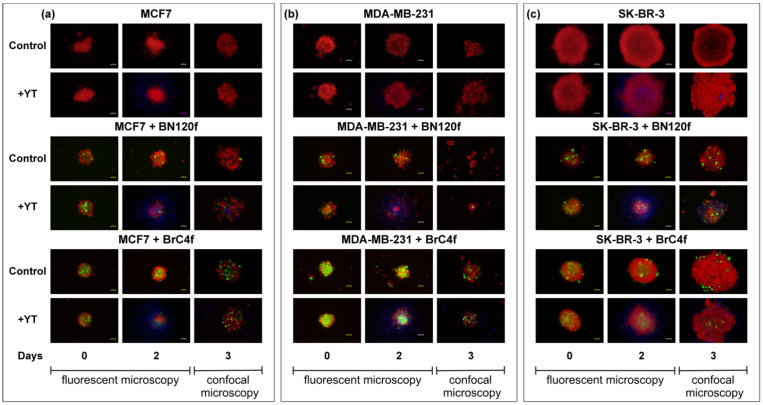
Localization of the NK-YT cells (labeled in blue) near the fibroblast-rich core from fibroblasts (labeled in green) in heterotypic cellular spheroids. Tumor cells labeled in red (**a**) MCF7; (**b**) MDA-MB-231; (**c**) SK-BR-3. Homotypic cellular spheroids from tumor as control. 0,2 days—fluorescent microscopy, 3 day—confocal microscopy.

**Figure 3 cells-14-01039-f003:**
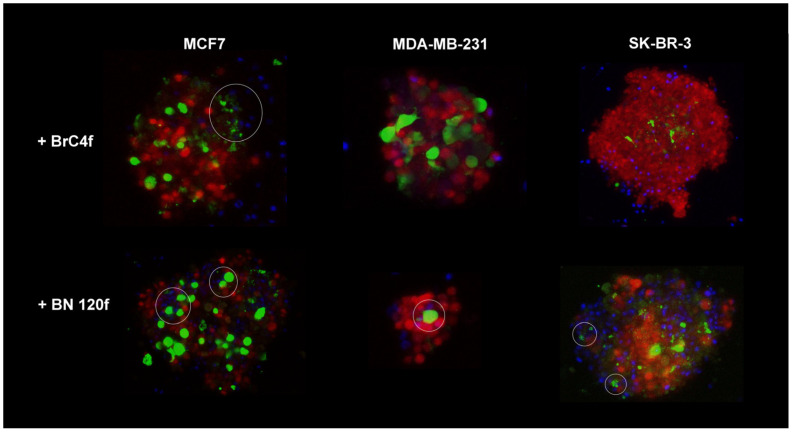
Cell-cell interactions in heterotypic cellular 3D model BC. Localization of the NK-YT cells (labeled in blue) near the fibroblast-rich core from fibroblasts (labeled in green). Tumor cells labeled in red. Confocal microscopy. Circles labeled interaction between stromal and immune cells.

**Figure 4 cells-14-01039-f004:**
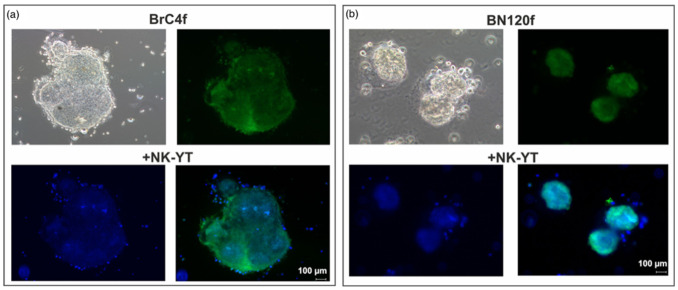
Interactions between stromal (green) and NK-YT (blue) cells in spheroid. Homotypic cellular spheroids from (**a**) cancer-associated fibroblast BrC4f; (**b**) normal fibroblast BN120f.

**Figure 5 cells-14-01039-f005:**
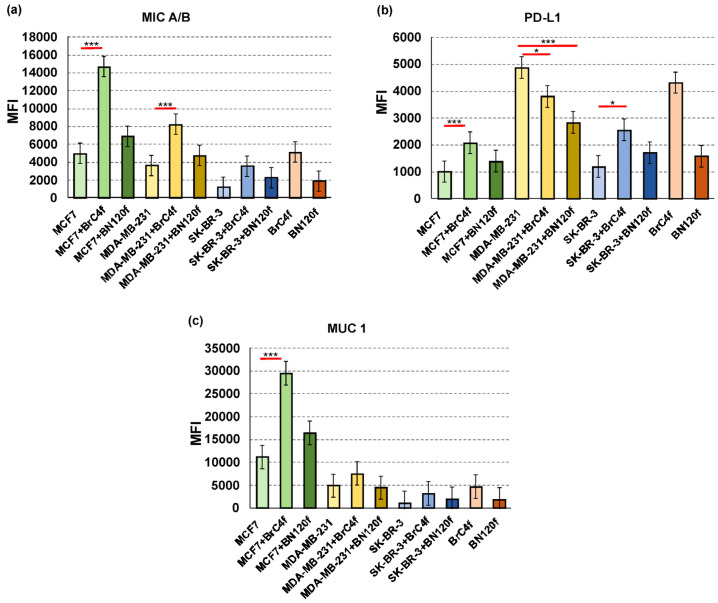
Flow cytometry is a technique used to analyze the expression, activation and inhibition of molecules located on the surface of cells after one day co-cultivation with YT. MFI values of (**a**) MIC A/B, (**b**) PD-L1 and (**c**) MUC1 were calculated for 3D and 3D-2. MFI—mean fluorescent intensity. * *p* < 0.05; *** *p* < 0.001 determined by unpaired two-tailed Student’s *t* test.

**Figure 6 cells-14-01039-f006:**
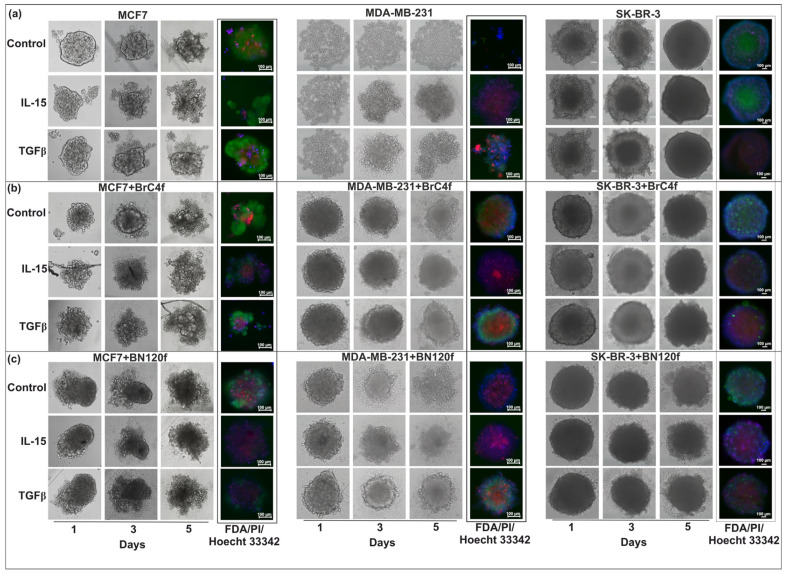
The dynamics of PB-NK cell infiltration into 3D- and 3D-2 models of breast cancer in the presence combination IL-2 (300 units/mL) with IL-15 (50 ng/mL) or TGFβ (5 ng/mL). Analysis of cell viability in the spheroid from (**a**) only tumor cells; (**b**) with CAFs; (**c**) with normal fibroblast. Green signal (FDA)—live cells, red (PI)—dead cells, blue (Hoechst 33342)—total number of cells. Monitoring of cells were observed during a 5-day co-culture period.

**Figure 7 cells-14-01039-f007:**
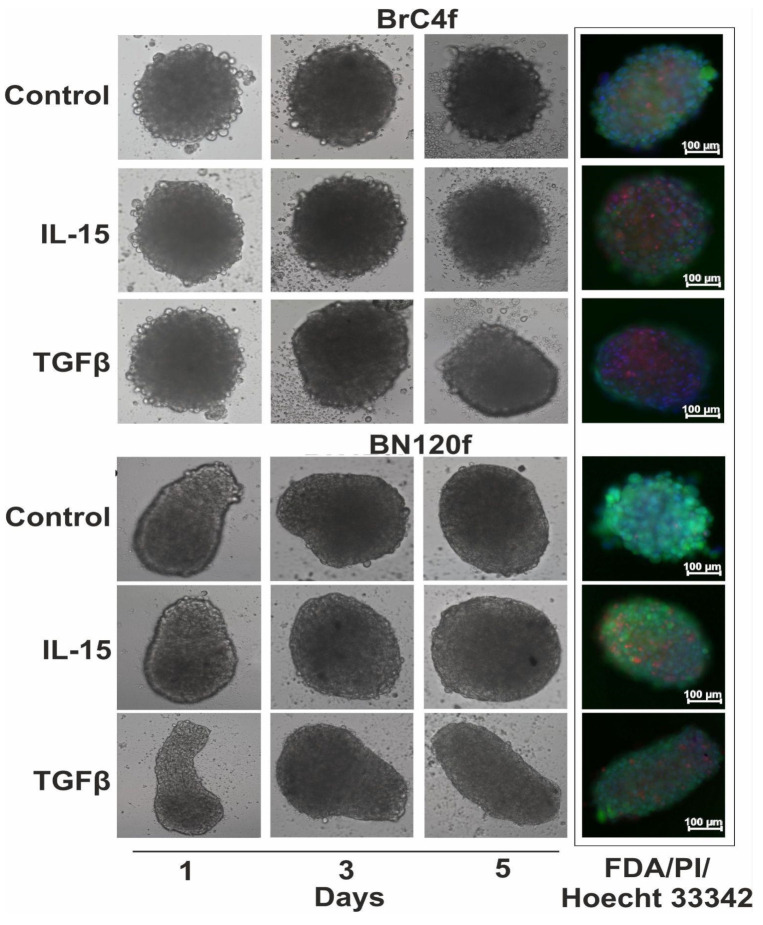
The dynamics of PB-NK cell infiltration into stromal spheroids cancer in the presence combination of IL-2 (300 units/mL) with IL-15 (50 ng/mL) or TGFβ (5 ng/mL). Green signal (FDA)—live cells, red (PI)—dead cells, blue (Hoechst 33342)—total number of cells. Monitoring of cells were observed during a 5-day co-culture period.

**Figure 8 cells-14-01039-f008:**
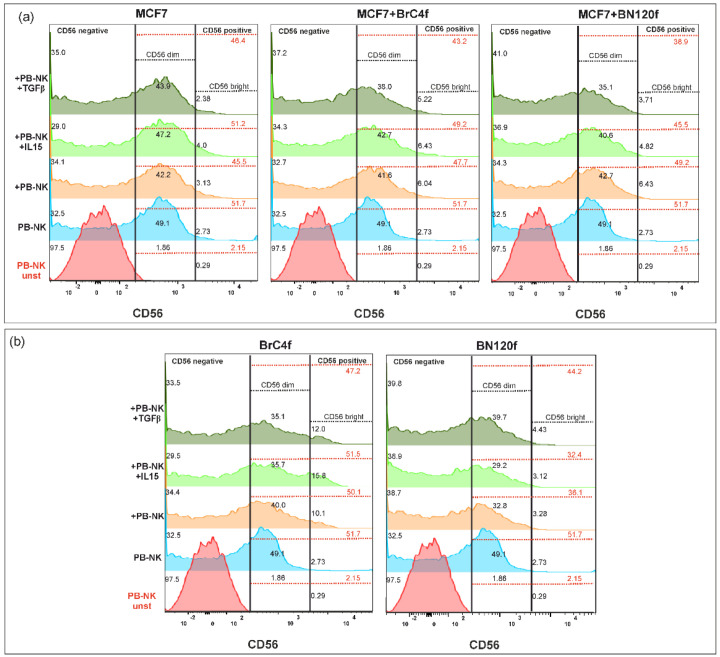
The expression of CD56 on PB-NK cells after co-cultivation with (**a**) MCF7 3D model BC cell-based system and (**b**) stomal 3D model. The color values of the histogram are as follows: red indicates the unstimulated control, blue indicates the control PB-NK cells, orange indicates PB-NK cells cultured without additional stimulation, light green indicates PB-NK cells cultured with IL-15, and dark green indicates PB-NK cells cultured with TGFβ.

**Figure 9 cells-14-01039-f009:**
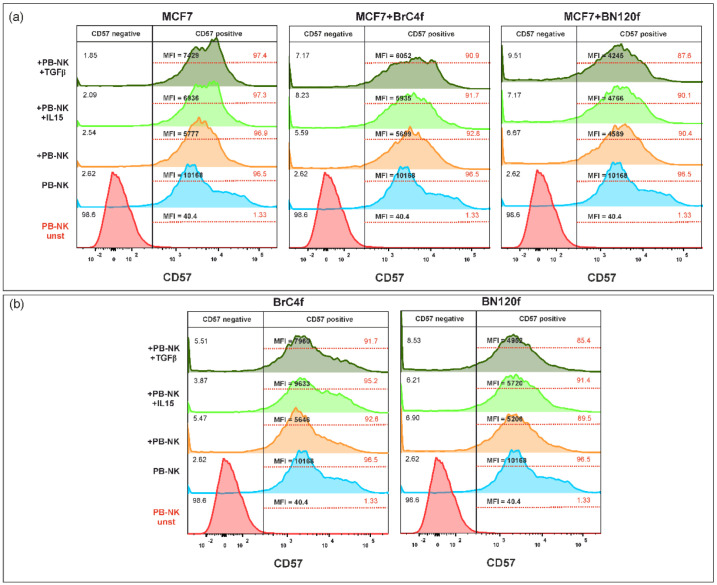
Phenotypic analysis of CD57 on PB-NK cells after co-cultivation with (**a**) MCF7 3D model BC cell-based system and (**b**) stromal 3D model. The colour values of the histogram are as follows: red indicates the unstimulated control, blue indicates the control PB-NK cells, orange indicates PB-NK cells cultured without additional stimulation, light green indicates PB-NK cells cultured with IL-15, and dark green indicates PB-NK cells cultured with TGFβ. MFI—mean fluorescent intensities.

**Figure 10 cells-14-01039-f010:**
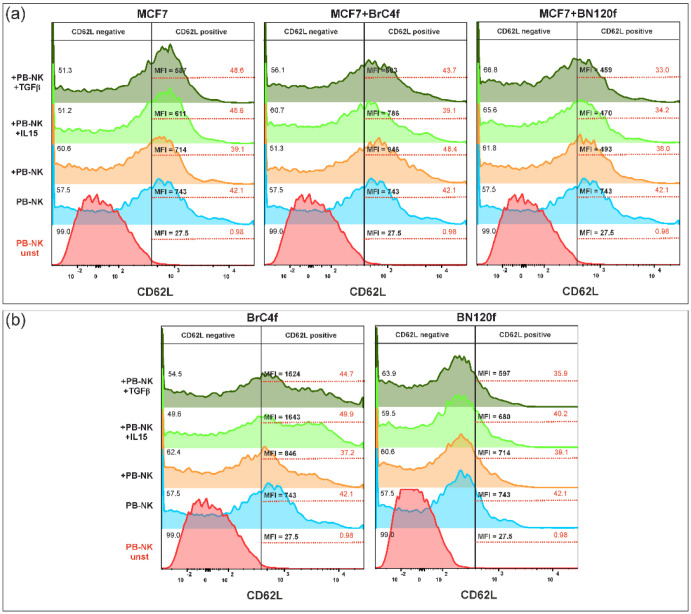
Phenotypic analysis of CD62L on PB-NK cells after co-cultivation with (**a**) MCF7 3D model BC cell-based system and (**b**) stomal 3D model. The color values of the histogram are as follows: red indicates the unstimulated control, blue indicates the control PB-NK cells, orange indicates PB-NK cells cultured without additional stimulation, light green indicates PB-NK cells cultured with IL-15, and dark green indicates PB-NK cells cultured with TGFβ. MFI—mean fluorescent intensities.

**Figure 11 cells-14-01039-f011:**
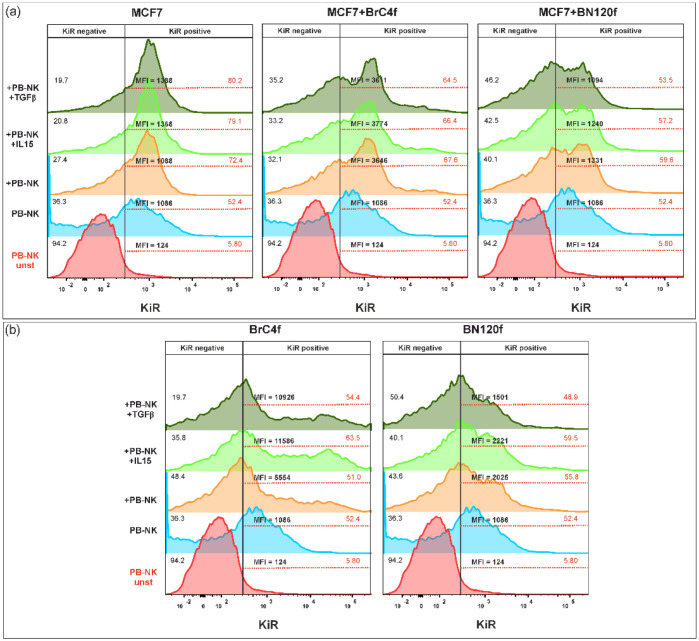
Phenotypic analysis of KIR on PB-NK cells after co-cultivation with (**a**) MCF7 3D model BC cell-based system and (**b**) stomal 3D model. The color values of the histogram are as follows: red indicates the unstimulated control, blue indicates the control PB-NK cells, orange indicates PB-NK cells cultured without additional stimulation, light green indicates PB-NK cells cultured with IL-15, and dark green indicates PB-NK cells cultured with TGFβ. MFI—mean fluorescent intensities.

## Data Availability

The data that support the findings of this study are available from the authors upon reasonable request.

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
