# Peer review of "The Activity of Human NK Cells Towards 3D Heterotypic Cellular Tumor Model of Breast Cancer"

_cells, 2025, doi:10.3390/cells14141039_

Round 1

Reviewer 1 Report

Comments and Suggestions for Authors

Reviewer comments

The manuscript titled “The activity of human NK cells towards heterotypic cellular 3D in vivo-like tumor model of breast cancer” by Leonteva et al. is a well-structured, timely, and engaging article. This study investigates the cytotoxicity of NK cells against breast cancer cells in a 3D model and examining the effect of cytokines IL-15 and TGFβ. I suggest several revisions to enhance clarity and strengthen the manuscript’s impact before publication.

Comments for Authors

  1. The authors are advised to refine the title of this manuscript, 3D is not consider as in vivo.

  1. The authors need to clarify, which particular signaling pathway involved in the effect on NK cell activity?

  1. Please justify for choosing specific breast cancer cells and fibroblast types? .

  1. Is there any specific reason for performing FDA staining before PI/Hoechst 33342 staining? Whether it affects the results?  Please explain.

  1. The authors could ensure that statistical analysis to evaluate the effects of IL-2, IL-5, and TGFβ on NK cell activity.

  1. If the authors having more clear image for fluorescence, I advise to replace.

  1. The authors should carefully check for grammatical errors and typos.

Author Response

Reviewer 1.

Comments for Reviewer 1:

Point 1. The authors are advised to refine the title of this manuscript, 3D is not consider as in vivo.

Response 1: We agree with this comment. We changed the title of this publication to «The activity of human NK cells towards 3D heterotypic cellular tumor model of breast cancer». Also, this phrase has been removed from the entire manuscript.

Point 2. The authors need to clarify, which particular signaling pathway involved in the effect on NK cell activity?

Response 2: We thank the reviewer for this valuable suggestion to identify an exact mechanism involved into NK-mediated activity. NK cell activation is mediated by different signals from multiple cell surface receptors; therefore, it could be challenging to establish the exact pathway responsible for NK cell activation in our test settings. The NKG2D receptor on natural killer (NK) cells plays a crucial role in their activation by initiating signaling pathways involving PI3K, PLCγ, and JAK2. This receptor might be involved in a process of NK activation which we show in our manuscript. Moreover, since antibodies are not involved in NK-cell activation in our model, we could assume, that CD16-dependent activation through CD3zeta/ZAP-70 is not the major pathway in our case. An assessment of downstream TGF-β signaling also identified TAK1-mediated activation of p38 MAPK as the critical pathway driving the conversion. IL-15 enhanced TGF-β-mediated conversion of NK-cells to an (ILC1)–like phenotype through Ras:RAC1 signaling as well as via the activation of MEK/ERK 10.4049/jimmunol.1900866. In the current version of the manuscript, we added it in our discussion (See line 792-794, 846-848).

Point 3. Please justify for choosing specific breast cancer cells and fibroblast types?

Response 3: In the current version of the manuscript, we discussed the selection of these particular cell cultures for spheroid formation (See lines 49-50, 115-118, 120-121, 482-483, 486-488, 798-801).

Point 4. Is there any specific reason for performing FDA staining before PI/Hoechst 33342 staining? Whether it affects the results?  Please explain.

Response 4: We thank the reviewer for these comments. The staining was performed according to 10.3390/cells9122557 with some modifications. The rationale underpinning the utilization of fluorescein diacetate staining prior to PI (propidium iodide) or Hoechst 33342 staining in cell viability assays is of significance due to the divergent mechanisms and cellular requirements of these dyes. It has been established that only live cells with intact membranes and active esterases can convert FDA into fluorescent fluorescein, which is then retained in viable cells 10.1177/33.1.2578146. PI is a membrane-impermeant dye that enters cells only when the cell membrane is compromised (i.e. when the cell is dead or dying). Hoechst 33342 is a DNA-binding dye that has a capacity to stain all nuclei (both live and dead cells) due to its ability to passively diffuse into cells. In the event of PI or Hoechst being added first, there is the potential for damage to cells or alteration of membrane permeability, which may result in false-negative FDA staining (reduced fluorescein signal). Moreover, PI has been observed to induce membrane disruption in certain curcumstances, thereby impacting FDA hydrolysis PMID: 20814586. Furthermore, it is imperative to note that the incubation times for FDA and PI/ Hoechst 33342 are not congruent, in order to circumvent erroneous results. Аll essential information regarding staining techniques has now been incorporated into the Methods Section (See line 246-256).

Point 5. The authors could ensure that statistical analysis to evaluate the effects of IL-2, IL-5, and TGFβ on NK cell activity.

 Response 5: A total of three independent experiments were conducted for the entire series of studies. We added this information in methods. Moreover, a statistical analysis of spheroid volumes in the experiment conducted with the potential for cytokine and PB-NK treatment (See Figure S10).

Point 6. If the authors having more clear image for fluorescence, I advise to replace.

Response 6: Additional images have been included, offering a visual exposition of confocal microscopy drawings. This exposition elucidates the phenomenon of cell-cell interactions in the context of a heterotypic model (See Fig.3, S5, S8).

Point 7. The authors should carefully check for grammatical errors and typos.

Response 7: Thank you for your kind consideration of our work. In the current version of the manuscript, we corrected minor spelling and typo errors. Current version of the manuscript was proofed by English speaker.

Reviewer 2 Report

Comments and Suggestions for Authors

The authors examined the cytotoxic activity of natural killer (NK) cells using 3D spheroid model using both breast cancer cells and fibroblasts from normal breast tissues and breast cancer tissues. Although some data was interesting, there are some points to be addressed.

  1. The morphology of 3D-2 models should be explained in more detail. Although the authors described that fibroblasts formed the core of spheroid which was surrounded by cancer cells, the morphology is unclear in Figure 2. Please provide pictures of confocal images of 3D-2 models.
  2. In addition, the formation of 3D2 models of present study (inner core consisting of stromal cells, surrounded by carcinoma cells) is a little bit strange because the localization of these cells seems reversed. Usually, the cancer cell cluster is surrounded by stromal components.
  3. Does live/dead determined by fluorescence really reflect the cytotoxicity of NK cells? The center of spheroids is sometimes under hypoxic conditions and this may affect the viability of cells in the center. As shown in Figure S2, dead cells are detected in the center of sphere.
  4. The authors demonstrated that the expression of MIC A/B, PD-L1, and MUC1 was altered in the 3D-2 spheroids (Figure 4) and these data was obtained from flowcytometry. The cells should have been resuspended before the assay. Please provide detailed information regarding the method of resuspension. In addition, in which cells did the authors detect the change of gene expression? Cancer cells, fibroblasts, or both?
  5. The findings from Figure 2 and Figure 3 should be confirmed using PB-NK cells.

Author Response

Reviewer 2.

We thank the reviewer for the positive feedback on our manuscript.

Comments for Reviewer2:

Point 1. The morphology of 3D-2 models should be explained in more detail. Although the authors described that fibroblasts formed the core of spheroid which was surrounded by cancer cells, the morphology is unclear in Figure 2. Please provide pictures of confocal images of 3D-2 models.

Response 1: We thank the reviewer for these valuable comments. We added the descriptive characterization of morphology described in our previous studies10.1134/S1990519X22060050 (See line 344-349, 703-710). In addition, Figure 3 (confocal microscopy) has been incorporated, which provides a detailed depiction of the localization of various cell types within the heterotypic model.

Point 2. In addition, the formation of 3D2 models of present study (inner core consisting of stromal cells, surrounded by carcinoma cells) is a little bit strange because the localization of these cells seems reversed. Usually, the cancer cell cluster is surrounded by stromal components.

Response 2: Spatial distribution has already been reported in former studies that focused on the development of co-culture spheroids comprising breast cancer cells and fibroblasts 10.1016/j.ejps.2023.10656010.1038/s41598-020-78087-7 Our results obtained serve to corroborate this tendency. Suggests that the stroma will become increasingly appreciated in the initiation or formation of breast carcinomas 10.4161/cam.20567 It is hypothesized that the secretion of extracellular matrix proteins is the primary factor contributing to the accelerated aggregation of stromal cells. This process enables the formation of a nucleus (central core of the spheroid), while cancer cells remain in the periphery. Please also see line 722, 726-729 where these data are discussed in our revised manuscript.

Point 3. Does live/dead determined by fluorescence really reflect the cytotoxicity of NK cells? The center of spheroids is sometimes under hypoxic conditions and this may affect the viability of cells in the center. As shown in Figure S2, dead cells are detected in the center of sphere.

Response 3: We thank the reviewer for these comments. Experiments involving the use of NK cells were conducted on the fifth day of spheroid formation. By the fifth day, the majority of cells in all types of spheroids are viable, and the formation of necrotic nuclei is not observed (See Figure S2). Moreover, сell viability staining was performed for the highest volume cell model (SK-BR-3). We showed that that after total 7 days of SK-BR-3 spheroid culturing most of the cells are still viable, indicating that metabolic processes do not impact NK cell activity (3 days co-cultivation) (See line 752-764, Figure S9). In addition, our revised manuscript includes a time-lapse demonstrating the efficacy of immune cells in eradicating a representative cell line (MDA-MB-231) (See Figure S5).

Point 4. The authors demonstrated that the expression of MIC A/B, PD-L1, and MUC1 was altered in the 3D-2 spheroids (Figure 4) and these data was obtained from flow cytometry. The cells should have been resuspended before the assay. Please provide detailed information regarding the method of resuspension. In addition, in which cells did the authors detect the change of gene expression? Cancer cells, fibroblasts, or both?

Response 4: In the current version of the manuscript, we added information about flow cytometry sample preparation (See line 283-300). The expression changes of MIC A/B, PD-L1, and MUC1 on the cell surface were analyzed using both 2D (See Figure S7), 3D models from tumor and stromal cells and heterotypic spheroids (Figure 5). Changes in the expression of proteins involved in the immune response were observed in heterotypic models that contained CAFs, as opposed to normal stromal cells. A comparison of the differences in expression levels between CAFs and normal fibroblasts cultured in 3D revealed that CAFs exhibited increased levels of MICA/B and PDL1. These data suggest that cancer-associated fibroblasts (CAFs) play a significant role in modulating the inflammatory response within heterotypic spheroid models. Changes in the expression of proteins involved in the immune response were observed in heterotypic models containing CAFs, in contrast to normal stromal cells. We have added additional discussion points in the revised manuscript (please see line 407-408, 419-426, 786-787, 800-801).

Point 5.  The findings from Figure 2 and Figure 3 should be confirmed using PB-NK cells.

Response 5: We would like to express our gratitude to the reviewer for this valuable suggestion to perform confocal microscopy with PB-NK. PB-NK cells and YT cells are both types of immune cells, but they have different origins and characteristics. PB-NK cells are a natural component of the innate immune system found in the blood, while YT is a cell line derived from a human T-cell leukemia tumor. It has been demonstrated that both cell types exhibit cytotoxic activity; however, the primary function of YT cells is as a research tool, whereas PB-NK cells have been the subject of more extensive study with regard to their therapeutic applications. As delineated in the introduction (See line 68-75), a comparative analysis of the distinguishing characteristics between NK cell models has been incorporated. This introduction has been augmented with elucidations that address the disparities among these models. As demonstrated in the accompanying Figure S6, PB-NK demonstrated high levels of activity in the elimination of tumor cells when administered at the same ratio as YT. Consequently, it is challenging to replicate such an experiment using confocal microscopy. Concurrently, it was observed that the incorporation of PB-NK into the spheroids at a reduced concentration resulted in the complete dissolution of the MDA-MB-231 spheroids. This precluded the preparation of the samples for confocal microscopy. Furthermore, it has been demonstrated that PB-NKs localize along the surface of spheroids for other cell types (see Figure 6 and 7, control panel). Nevertheless, consensus has been reached that it is more appropriate to continue working with penetration of PB-NK into spheroids. In the subsequent study, experiments will be planned to address this issue. However, it should be noted that it is not possible to promptly isolate fresh PB-NK for use, as this process requires additional time and modification preparation of the samples for confocal microscopy.

Round 2

Reviewer 2 Report

Comments and Suggestions for Authors

The manuscript is well revised.